# MetaGS: A Meta-Learned Gaussian-Phong Model for Out-of-Distribution 3D Scene Relighting

**Yumeng He**    **Yunbo Wang***
MoE Key Lab of Artificial Intelligence, AI Institute, School of Computer Science
Shanghai Jiao Tong University
{ymhe, yunbow}@sjtu.edu.cn

## Abstract

Out-of-distribution (OOD) 3D relighting requires novel view synthesis under unseen lighting conditions that differ significantly from the observed images. Existing relighting methods, which assume consistent light source distributions between training and testing, often degrade in OOD scenarios. We introduce **MetaGS** to tackle this challenge from two perspectives. First, we propose a meta-learning approach to train 3D Gaussian splatting, which explicitly promotes learning generalizable Gaussian geometries and appearance attributes across diverse lighting conditions, even with biased training data. Second, we embed fundamental physical priors from the *Blinn-Phong* reflection model into Gaussian splatting, which enhances the decoupling of shading components and leads to more accurate 3D scene reconstruction. Results on both synthetic and real-world datasets demonstrate the effectiveness of MetaGS in challenging OOD relighting tasks, supporting efficient point-light relighting and generalizing well to unseen environment lighting maps.

## 1   Introduction

3D scene relighting generates novel lighting effects that interact with the observed 3D environment, with recent advances in learning-based volume rendering offering effective solutions [21, 36, 23, 40, 24, 22, 3, 30]. A typical approach captures a substantial number of multi-view images of a scene under individual lighting conditions, then trains a NeRF or 3D Gaussian splatting (3DGS) model to generalize to new point light positions [28, 13, 9, 41, 8]. Notably, such idealized data requirements are often impractical in real-world scenarios. To simulate a more realistic capturing process, we build on prior work using a **One Light At a Time (OLAT)** setup [34, 14, 1], where training data is collected with a moving point light and a moving camera. Recent studies have further explored low-cost OLAT capture using a smartphone flashlight as the moving light source [5].

While most OLAT methods assume a coherent lighting distribution between training and testing, real-world scenarios often involve light sources that are randomly distributed around the scene, which may result in biased training data. This leads to an **out-of-distribution (OOD)** issue where test-time light sources deviate from the training distribution. This OOD relighting presents a more natural and challenging task, serving as a robust test for evaluating a model's performance under "*truly*" novel lighting conditions. Compared with the standard OLAT setup, it complicates the learning of true object illumination properties and geometries due to a limited training corpus with entangled, time-varying lighting and viewing directions and, more critically, spatially biased lighting. As shown in Figure 1, existing OLAT methods struggle with the OOD relighting task, often producing unrealistic lighting effects, such as chaotic specular highlights and shadows, due to overfitting to the training samples. Also, because of the lack of generalizability, when new lighting distributions are introduced, they typically require full retraining or finetuning to adapt.

---

*Corresponding author.

39th Conference on Neural Information Processing Systems (NeurIPS 2025).

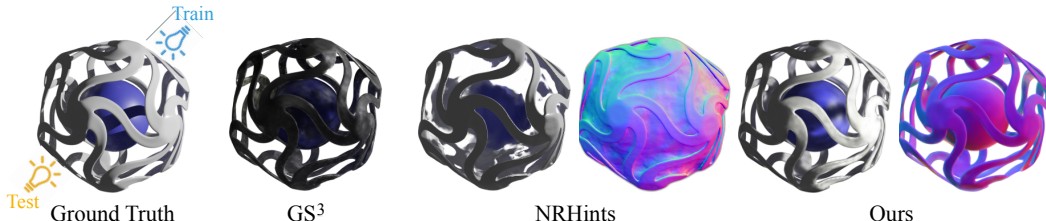

Train

Test    Ground Truth          GS³              NRHints              Ours

Figure 1: *Preliminary results.* NRHints [34] and GS³ [1] struggle with out-of-distribution light positions (as specified in Section 2), as the blurry novel view synthesis results and erroneous normal predictions indicate. GS³ does not require well-defined surface normals and does not directly output the normal image.

In this paper, we present two novel techniques to enable better OOD relighting. First, we provide a pilot study on **meta-learning-based 3DGS** and introduce MetaGS to mitigate overfitting and improve the generalizability of the OLAT models. This approach leverages bilevel optimization techniques to effectively address the challenges associated with limited training data and complex lighting variations. Specifically, we frame OLAT as a multi-task learning problem, treating rendering under specific lighting positions as distinct tasks. The core idea is to validate the learned Gaussian illumination properties under a given lighting condition using data from other lighting conditions. The resulting second-order gradients of 3DGS parameters can explicitly encourage the model to generalize to unseen illumination rather than overfitting to specific training samples, as is common with standard 3DGS loss functions.

Furthermore, MetaGS integrates simple yet fundamental physical priors into Gaussian splatting by incorporating **a learnable Blinn-Phong reflection model** [2]. This approach effectively decouples different shading components—diffuse, specular, and ambient—leading to a more physically grounded understanding of the interactions between objects and lighting in the training data. This design draws inspiration from prior research [39, 28, 13] indicating that modular and compositional representations enhance generalization capabilities. The decoupled rendering improves the model's ability to generalize to novel lighting conditions, allowing it to independently adjust each shading component according to the specific lighting scenario.

We evaluate MetaGS on both synthetic and real-world datasets. It supports efficient point-light relighting under highly constrained training illuminations and significantly outperforms existing OLAT methods in OOD relighting tasks. We further demonstrate that both meta-learning and the differentiable Phong model individually contribute to improved generalization. Notably, MetaGS generalizes well to unseen environment maps, despite being trained exclusively in OLAT scenarios.

## 2   Preliminaries

**Definition and challenges of OLAT.**    OLAT refers to a specialized setting for 3D reconstruction and relighting. This involves illuminating the subject with a single point light in sequential exposures, capturing one image at each light position. Each capture is represented as a tuple $\{O_t, V_t, P_t\}$, where $O_t$ denotes the observed image, $V_t$ represents the camera configuration, and $P_t$ contains lighting information. In our context, $P_t$ specifies the position of the point light. Unlike the "*multiple lights, multiple cameras*" setup, which requires multi-view images captured by multiple time-synchronized cameras for each lighting condition, OLAT relies on a single camera. This significantly simplifies data collection, particularly in scenarios with dynamic lighting. The rapidly changing light positions in OLAT introduce a new challenge for 3D reconstruction. The color of a 3D point can vary significantly, increasing ambiguity for estimating true object geometries and illumination properties.

**Preliminary findings.**    We summarize existing OLAT relighting methods in Table 1. In the preliminary experiments, we evaluate NRHints [34], the prior art in NeRF-based relighting models. We observe that it underperforms in testing scenarios with out-of-distribution (OOD) lighting positions, a feature that we believe should be crucial for relighting methods. Specifically, in Figure 1, the lighting in the training set is arranged on one side of the hemisphere, while the lighting in the test set is arranged on the opposite side (cameras are located on both sides). In such cases, NRHints fails to generate reasonable rendering results, likely due to its implicit modeling of shadows and

Table 1: Comparison of the OLAT relighting methods.

|  | [35] | [5] | [6] | [34] | [1] | [12] | Ours |
|---|---|---|---|---|---|---|---|
| Point light | ✓ | ✓ | ✓ | ✓ | ✓ | ✓ | ✓ |
| Shadow computation | ✗ | ✗ | ✓ | ✓ | ✓ | ✓ | ✓ |
| OOD light | ✗ | ✗ | ✗ | ✗ | ✗ | ✗ | ✓ |

specular reflections. We additionally evaluate a concurrent 3DGS-based approach [1] and observe similar degraded results under the OOD relighting setup. These findings highlight that understanding intrinsic illumination properties under arbitrary lighting variations remains a challenging task, as the model tends to overfit to perspective-constrained observations and fail to leverage enough physical principles when dealing with unseen lighting distribution.

## 3  Method

In this section, we present the details of MetaGS for addressing the OOD OLAT learning challenge:

- In Section 3.1, we incorporate physical shading priors within Gaussian splatting via a differentiable Phong reflection model to decouple mixed illumination components.

- In Section 3.2, we introduce a bilevel optimization scheme to estimate light-independent scene geometries and intrinsic illumination properties, marking an early effort in volume rendering.

- In Section 3.3, we discuss the entire training pipeline and the implementation details.

### 3.1  Differentiable Phong Model

Our *MetaGS* method leverages simple yet generalizable physical priors from the Blinn-Phong model [19], which captures three fundamental components of light transport: ambient, diffuse, and specular reflections. The core idea is to disentangle these illumination components by learning the interactions between (i) the normal vectors of the Gaussian points, (ii) the viewing directions, and (iii) the ray direction from the point light.

Specifically, the ambient component represents constant environmental illumination, simulating indirect scattering from surrounding surfaces to establish a baseline brightness level. The diffuse component, based on Lambertian law, describes light scattering in multiple directions from rough surfaces. The specular reflection is computed based on the angle between the light direction and the bisector ($\mathbf{h}$) between the viewing direction ($\mathbf{v}$) and the light direction ($\mathbf{l}$), with a shininess exponent that represents different degrees of glossiness.

In MetaGS, we extend the original 3DGS by introducing the decoupled computation for different shading components. As shown in Figure 2, in addition to the basic Gaussian attributes (*e.g.*, position $\mathbf{x}$, rotation $R$, scale $S$, opacity $\alpha$, and spherical harmonic coefficients $f$), each Gaussian point is further associated with a normal vector $\mathbf{n}$, a 3-channel diffuse color $k_d$, and a 1-channel specular coefficient $k_s$. The newly added attributes enhance the model's understanding of lighting effects, facilitating the learning process of light-independent object geometries. The overall color of a Gaussian point is determined by:

$$L_p = L_a + L_d + L_s = L_a + \sum_{|\text{lights}|} (k_d I_d + k_s I_s), \tag{1}$$

where $k_{\{d,s\}}$ and $I_{\{d,s\}}$ are the colors and intensities of the diffuse and specular components. $|\text{lights}|=$ 1 in OLAT. In graphics, zero-order spherical harmonics (SH) coefficients typically represent the basic, uniform component of a function defined over the sphere, essentially the average or constant part of a lighting environment. Therefore, we restrict Gaussians' SH coefficients to zero order, denoted by $f_0$, corresponding to the ambient color. We compute the diffuse and specular light transports by multiplying the color and intensity of each component, where the intensities $I_d$ and $I_s$ are defined as:

$$I_d = \frac{I}{r^2} \max(0, \mathbf{n} \cdot \mathbf{l}), \quad I_s = \frac{I}{r^2} \max(0, \mathbf{n} \cdot \mathbf{h})^p, \tag{2}$$

where $I$ denotes the light emitted intensity, which is a global learnable parameter, $\mathbf{l}$ denotes the point-to-light normalized vector, and $\mathbf{h} = \frac{\mathbf{v}+\mathbf{l}}{||\mathbf{v}+\mathbf{l}||}$ denotes the bisector of the point-to-camera normalized

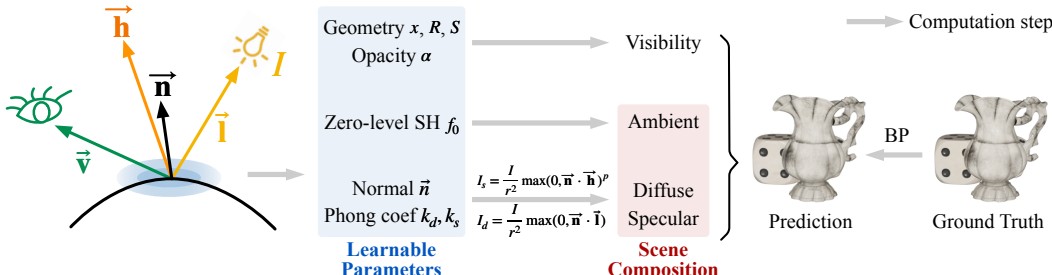

Figure 2: *The model design of MetaGS.* Our model decomposes the illumination effects by interacting the learned Gaussian points with rays originating from both the viewer and the light source.

vector $\mathbf{v}$ and $\mathbf{l}$. We model the coefficients $k_d$ and $k_s$ implicitly, where $k_s$ is multiplied by the RGB color of the point light.

**Shadow computation.** We calculate shadow effects using a BVH-based ray tracing method [8]. Similar to the camera-to-point accumulated transmittance, the received light intensity $T_i^{\text{light}}$ represents the total transmittance of light along the light-to-point ray. It is therefore affected by the opacity of the surfaces encountered along this path. For each point, we determine light visibility by tracing a ray from the Gaussian's center to the light source. Since only the diffuse and specular colors are influenced by incident light intensity, we update the color of each point by incorporating the light visibility factor into the diffuse and specular terms from Eq. (1):

$$L_p = L_a + T_i^{\text{light}} \sum_{|\text{lights}|} (k_d I_d + k_s I_s). \tag{3}$$

This explicit formulation accounts for physics-based shadowing effects, provides strong interpretability, and improves generalizability compared to the implicit, high-dimensional shadow modeling methods. However, integrating this module directly into MetaGS's pipeline presents challenges. Under OLAT conditions, learning coherent geometry is challenging, leading to difficulties in accurately separating shadows from the scene. This, in turn, results in erroneous color predictions. The following meta-learning scheme promotes the mutual learning of scene geometry and appearance, alleviating the difficulties in learning coherent attributes under varying illuminations.

### 3.2 Meta-Learned Gaussian Relighting

Existing 3D relighting methods exhibit performance degradation when handling out-of-distribution relighting, primarily due to overfitting lighting patterns to perspective-constrained observations, resulting in producing unreasonable lighting components (such as wrong specular and shadows).

To address this, we introduce a meta-learning framework based on bilevel optimization, which has been shown to effectively bridge the distribution shift between the training and testing domains, facilitating the generalization of optimized variables to unseen scenarios [4]. This training strategy can also promote coherent geometry learning in complex OLAT tasks, as shown in the ablation study. In MetaGS, the intuition of incorporating bilevel gradient update is to mitigate overfitting to specific light conditions by explicitly simulating test samples with OOD light sources during each gradient update, thereby improving the model's ability to generalize to varied lighting scenarios.

As illustrated in Alg. 1, we organize the training processes with different $\{O_t, V_t, P_t\}$ as multiple learning tasks. We alternate model training between these tasks, validating the optimized variables under one lighting condition using data sampled from other conditions. This learning procedure encourages both the lighting attributes $(f, k_a, k_s)$ and the geometric attributes $(\mathbf{x}, \mathbf{n}, R, S, \alpha)$ of the Gaussian points to converge cohesively.

To simulate test-time conditions, the training data is divided into support (*training*) and query (*validation*) sets. At each iteration, $2m$ samples are drawn from the training data to form the support set $\mathcal{D}_{1:m}^{\text{sup}}$ and query set $\mathcal{D}_{1:m}^{\text{query}}$ (Line 5). Each support sample pairs with a query sample, denoted by subscript $i$. During each training step, the model alternates between these tasks in the inner optimization loop (Lines 6-9) and the outer loop (Line 10).

---

**Algorithm 1** Meta-Training for OOD Relighting

---
1: **Input:** Training set $\{\mathcal{D}_t\}_{1:T}$, where $D_t = \{$light position, camera parameters, RGB image$\}$
2: **Hyperparameters:** Learning rates $\alpha, \beta$
3: **Initialized parameters:** Pretrained Gaussian attributes $\{\theta_k\}$, where $\theta_k = (\mathbf{x}, \mathbf{n}, R, S, \alpha, f_0, k_d, k_s)$
4: **while** not converge **do**
5:     Sample disjoint data $\mathcal{D}_{1:m}^{\text{sup}}$ and $\mathcal{D}_{1:m}^{\text{query}}$ from $\{\mathcal{D}_t\}_{1:T}$
6:     **for** task $i$ in $\{1 : m\}$ **do**                             ▷ `Inner optimization loop`
7:         $\theta_i' \leftarrow \theta - \alpha \nabla_\theta \mathcal{L}_{\text{final}}(\theta; \mathcal{D}_i^{\text{sup}})$
8:     **end for**
9:     $\{\theta\} \leftarrow \{\theta\} - \beta \sum_{i=1}^{m} \nabla_\theta \mathcal{L}_{\text{final}}(\theta_i'; \mathcal{D}_i^{\text{query}})$            ▷ `Outer optimization step`
10: **end while**

---

**Inner optimization loop.** In the inner loop, the model is trained on each support set to independently generate $m$ sub-models, representing the hypothetical estimates $\theta_i'$ under each lighting condition. All sub-models start with the same parameters of $\theta$. The light intensity $I$ is a global parameter that is also learnable and optimized jointly with the Gaussian attributes. It is omitted here for clarity. We specify the loss function $\mathcal{L}_{\text{final}}$ in subsequent Section 3.3.

**Outer optimization loop.** In the outer loop, a global gradient update is performed across all sampled tasks, updating the Gaussian attributes. We first compute the loss function for each query sample, $\mathcal{L}_{\text{final}}(\cdot, \cdot; \mathcal{D}_i^{\text{sup}})$, using the corresponding inner-loop model hypotheses $\theta_i'$. We then aggregate the $m$ losses to update the initial $\theta$. This approach, involving task-specific adaptation followed by a global update, enables the model to learn generalizable representations across varying lighting conditions.

### 3.3 Entire Training Pipeline

We employ a three-stage training scheme. The first stage, similar to 3DGS, focuses on training the core Gaussian attributes, including position $\mathbf{x}$, rotation $R$, scale $S$, opacity $\alpha$, and spherical harmonic coefficients $f$. After this stage, the model tends to capture a rough geometry and an average color across different illuminations, serving as an effective initialization for further refinement. In the second stage, we incorporate the normal attributes into the optimization process, acknowledging that learning a Phong model heavily relies on accurate normal estimations. In the final stage, we integrate the diffuse and specular components, training all parameters concurrently through meta-learning.

**Objective functions.** We follow 3DGS [10] to compute the deviation of the predicted image from the ground truth, using an L1 regularization and a D-SSIM term as the RGB loss. We also employ a sparse loss [9] to encourage the opacity values $\alpha$ of the Gaussian spheres to approach either 0 or 1, thereby facilitating learning with opaque objects. For normal estimation, we follow GaussianShader [9] to compute the normal vectors using Gaussian's shortest axis direction, and progressively align them with the depth-inferred pseudo-normals.

**Implementation details.** We implement MetaGS using PyTorch [18]. For optimization, we use the Adam optimizer [11] with the same parameters as those specified by [10]. Our training procedure contains three stages: the first stage last for $10k$ iterations, the second stage last for $5k$ iterations, followed by the final meta-learning stage of $10k$ iterations. All experiments are conducted on a single NVIDIA RTX 3090 GPU.

## 4 Experiments

### 4.1 Experimental Settings

We perform three types of OLAT relighting experiments:

- *OOD relighting:* We introduce a novel view synthesis setup to assess the generalizability of 3D relighting models under OOD lighting conditions. The point lights in the training set are positioned on one side of the upper hemisphere, while the lighting in the test set is placed on the opposite side.

- *Camera-light-colocated relighting:* This setup mirrors the one used in IRON [35], where the camera and point light are co-located for each image. Similar to the OOD setting, the colocated

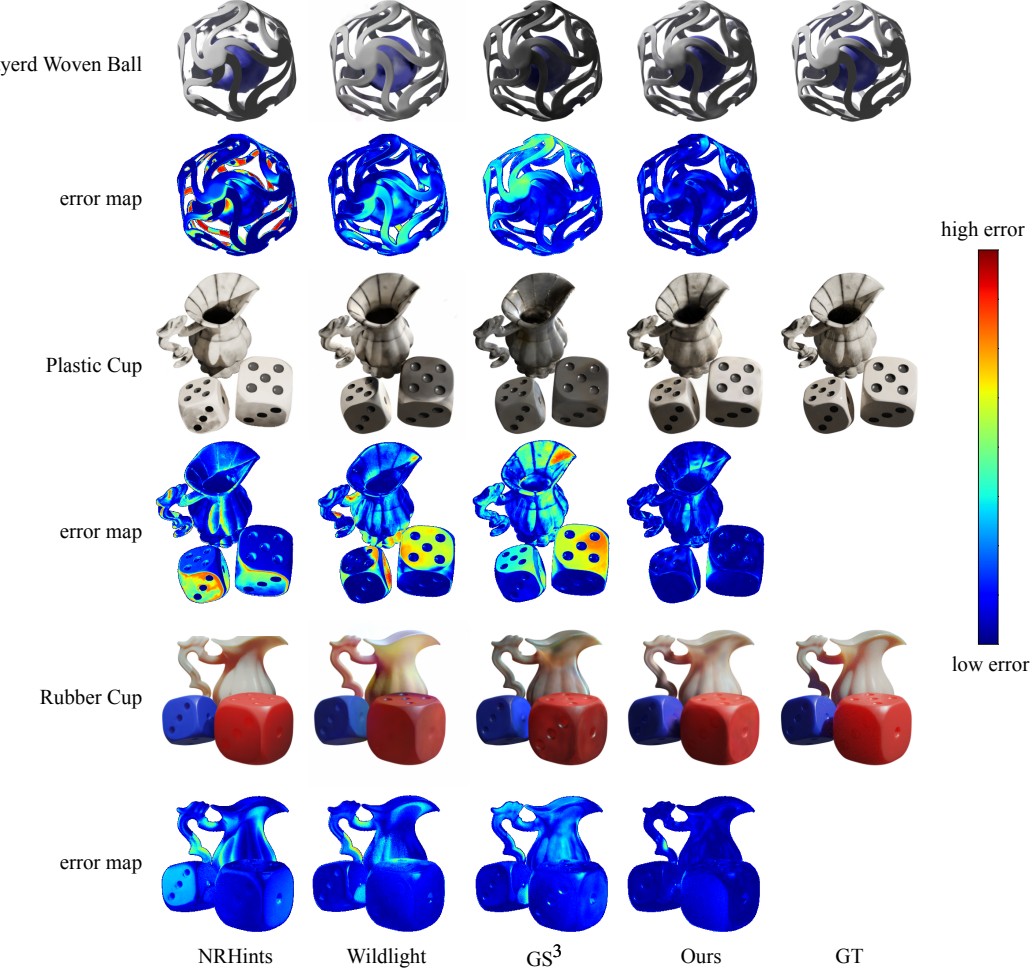

Figure 3: *OOD relighting results on synthetic data.* We present rendered novel views and error maps. While baselines often misrepresent shadows or light-dependent effects (*e.g.*, incorrect shadows in *Plastic Cup*), our model better infers surface appearance.

Table 2: *PSNR results for OOD relighting.* See our Appendix for full comparisons.

| Method | Synthetic | | | Real-world | | | | | | |
|---|---|---|---|---|---|---|---|---|---|---|
| | Ball | PlasCup | RubCup | Cat | Catsmall | CupFabric | Fish | FurScene | Pikachu | Pixiu |
| NRHints [34] | 17.25 | 23.92 | 27.44 | 18.04 | 24.63 | 24.65 | 22.57 | 21.55 | 24.00 | 23.03 |
| WildLight [5] | 21.73 | 20.95 | 24.02 | 18.65 | 22.53 | 24.03 | 21.47 | 20.33 | 19.09 | 20.22 |
| GS$^3$ [1] | 18.84 | 20.30 | 24.37 | 17.66 | 23.34 | 25.04 | 21.12 | 17.34 | 24.11 | 19.63 |
| Ours | **26.76** | **27.54** | **27.95** | **26.45** | **26.44** | **27.29** | **24.68** | **24.82** | **25.54** | **25.65** |

configuration also represents a constrained lighting regime, as the light source is fixed to the camera pose. For fair comparison, we generate colocated versions of the synthetic scenes accordingly.

- *Environment map relighting:* We evaluate the generalization of the models exclusively trained in the OLAT setup to novel view synthesis with unseen environment lighting maps.

We evaluate MetaGS on 3 synthetic scenes and 7 real captured scenes from NRHints [34], featuring a diverse range of materials, complex object shapes, significant self-occlusion, and intricate shadow effects. Each image is rendered (or captured) with a unique camera pose and point-light position. Full details of the datasets are provided in the Appendix.

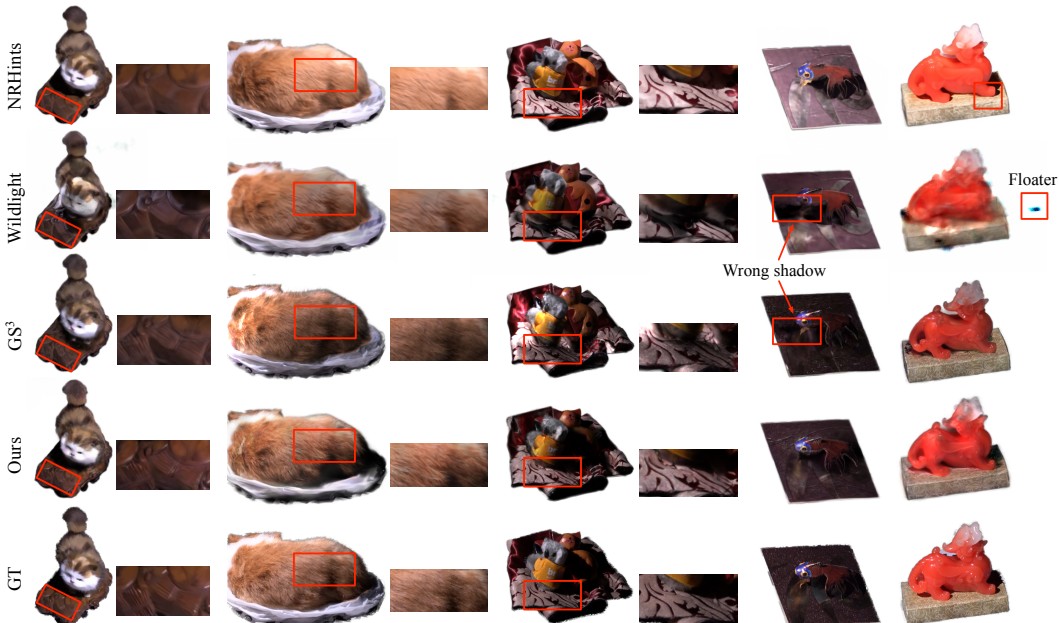

Figure 4: *OOD relighting results on real-world data.* As highlighted with the red boxes, baseline models struggle with some level of global shading consistency, including color shifts, incorrect shadows, and floating artifacts. Our approach presents physically plausible specular highlights and geometrically consistent shadows that closely match ground truths.

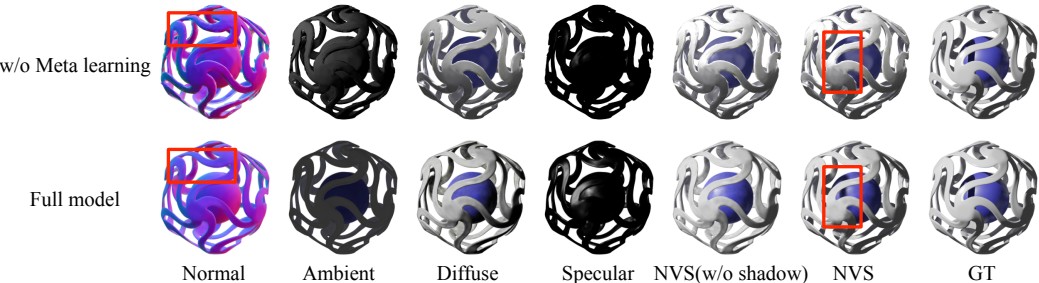

Figure 5: *Ablation studies with qualitative results.* As shown in the decoupling components, the meta-learning scheme is essential for learning object illumination properties and geometries. Without it, the model generates plain lighting effects, particularly for the diffuse and specular components.

For each synthetic scene, we generate a total of 600 OLAT images using Blender, with 500 for training and 100 for testing, following out-of-distribution or colocated data patterns. For the real-world scenes, we split the data according to their point light positions to construct out-of-distribution datasets, and use 600 training images per scene—a significantly smaller subset than the full NRHints dataset.

We compare MetaGS against both NeRF-based and Gaussian-based models, with a particular focus on the OOD relighting setup. The baseline models include state-of-the-art approaches from the past two years: GS$^3$ [1], NRHints [34], WildLight [5], Relightable 3DGS [8], GaussianShader [9], and IRON [35]. The evaluation metrics include PSNR, SSIM [25], and LPIPS [38].

## 4.2 Results for Out-of-distribution OLAT Relighting

We evaluate MetaGS in scenarios with OOD point light positions to explore its generalizability to unseen illuminations. The qualitative and quantitative results are respectively presented in Figure 3, Figure 4, and Table 2. Our model remarkably outperforms the baselines in handling complex light interactions. For example, in Figure 4, the lighting and shading effects on the cat's fur are rendered with high fidelity, while baseline models tend to produce incorrect color tones. Furthermore, as indicated by the error maps in Figure 3, all of three OLAT baselines tend to provide unrealistic results,

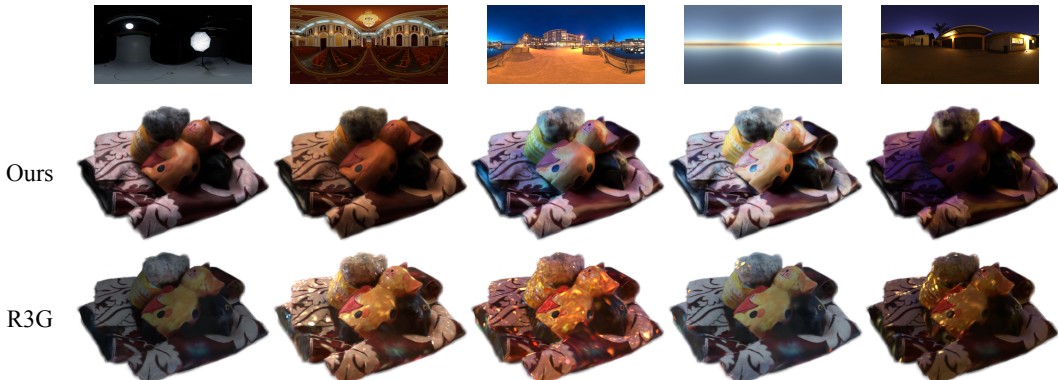

Figure 6: *Relighting with environment maps:* Our method, trained under OLAT settings, successfully generalizes to unseen environmental lighting; while the compared method is trained in an all-light-on setup that provides multiple lighting conditions per training view, it still fails to generalize to perform relighting with novel environment maps and exhibits visible artifacts.

especially for regions with significant specular reflections or shadows. Please see our Appendix for video visualizations on OOD relighting, where baselines' performance drops as the light enters the OOD region, while our model provides better relighting results.

The generalizability of MetaGS to OOD illumination stems from two key factors. First, it is due to the Phong model's ability to capture the underlying principles of light interaction. For example, instead of simply viewing specular highlights as the view-dependent color of a point (which is not generalizable), our method explicitly calculates the interactions of rays and surface normals, enabling the model to generate accurate highlights even in areas not directly illuminated during training. Second, meta-learning further enhances generalization by simulating test-time conditions during training, effectively reducing overfitting to specific illuminations. In the Appendix, we empirically show that the benefit of meta-learning does not come from a larger batch size.

**Ablation studies.** We conduct a series of ablation studies to validate the effectiveness of the meta-learning scheme as well as the shadow computing. From Table 3 and Figure 5, we observe that the proposed meta-learning training scheme strongly impacts performance. Without it, the model struggles to converge smoothly and make unrealistic component estimations, resulting in plain rendering quality.

Table 3: *Ablation studies of each model component for OOD relighting.* We show the average results of all three synthetic scenes.

| Method | PSNR$^\uparrow$ | SSIM$^\uparrow$ | LPIPS$^\downarrow$ |
|---|---|---|---|
| Full model | **27.42** | **0.9546** | **0.0505** |
| w/o Meta-learning | 19.14 | 0.8781 | 0.0892 |
| w/o Shadow | 21.53 | 0.9105 | 0.0735 |

### 4.3 Generalization to Other Relighting Setups with Constrained OLAT Training

**Camera-light-colocated relighting.** We evaluate MetaGS in the camera-light-colocated setup, as introduced by IRON [35]. Like the OOD scenario, the colocated setup also imposes limited lighting diversity, as the point light is tied to the camera's position throughout training. Quantitative results in Table 4 show that MetaGS outperforms IRON, achieving more accurate illumination inference under new point positions, and generating fewer rendering artifacts. Visualizations are presented in the Appendix.

Table 4: *Novel view synthesis results in PSNR under camera-light-colocated relighting setup.* Full results in all evaluation metrics are included in the Appendix.

| Method | Ball | PlaCup | RubCup |
|---|---|---|---|
| IRON [35] | 26.99 | 34.43 | 36.22 |
| Ours | **38.72** | **36.90** | **38.89** |

**Few-light and Fixed-elevation-light Relighting.** To evaluate MetaGS under constrained illumination conditions, we additionally design the following settings that vary in light distribution, coverage, and sparsity, simulating more realistic capture scenarios. In the few-light setting, only three training lights are sampled, each paired with 200 camera views in a one-light–multi-camera configuration. Two variants are considered: (1) *interpolation*, where training lights are uniformly distributed over

Table 5: Novel view synthesis results in the **few-light** and **fixed-elevation** relighting setup.

| Method | Fixed-elevation | | | Few lights (Interp) | | | Few lights (Extrap) | | |
|---|---|---|---|---|---|---|---|---|---|
| | PSNR | SSIM | LPIPS | PSNR | SSIM | LPIPS | PSNR | SSIM | LPIPS |
| NRHints [34] | 25.76 | 0.9633 | 0.0402 | 18.05 | 0.8823 | 0.0911 | 18.62 | 0.8960 | 0.0681 |
| GS$^3$ [1] | 22.93 | 0.9393 | 0.0580 | 19.92 | 0.9165 | /0.0693 | 18.06 | 0.8781 | 0.0920 |
| Ours | **27.48** | **0.9590** | **0.0371** | **24.90** | **0.9476** | **0.0462** | **23.94** | **0.9369** | **0.0533** |

Table 6: *Results of the proposed meta-learning scheme integrated with various relighting models.* PSNR values demonstrate the generalization performance, with $\Delta$ (Meta–Base) indicating the improvement from applying meta-learning.

| Method | Ball | PlasCup | RubCup | Cat | Catsmall | CupFabric | Fish | FurScene | Pikachu | Pixiu | Avg. | $\Delta$ |
|---|---|---|---|---|---|---|---|---|---|---|---|---|
| Ours | **26.76** | **27.54** | 27.95 | **26.45** | **26.44** | **27.29** | **24.68** | **24.82** | **25.54** | **25.65** | **26.31** | – |
| NRHint | 17.25 | 23.92 | 27.44 | 18.04 | 24.63 | 24.65 | 22.57 | 21.55 | 24.00 | 23.03 | 22.71 | – |
| +meta | 17.70 | 24.57 | **28.18** | 19.75 | 25.44 | 25.83 | 24.04 | 21.90 | 24.51 | 24.18 | 23.61 | 0.90 |
| RNG | 20.22 | 22.72 | 24.94 | NaN | 24.20 | 25.09 | NaN | 20.81 | 23.55 | NaN | 23.08 | – |
| +meta | 22.65 | 23.93 | 25.88 | 21.94 | 25.07 | 26.52 | 23.66 | 21.75 | 24.90 | 22.75 | 23.91 | 1.31 |
| GS$^3$ | 18.84 | 20.30 | 24.37 | 17.66 | 23.34 | 25.04 | 21.12 | 17.34 | 24.11 | 19.63 | 21.18 | – |
| +meta | 21.54 | 23.18 | 26.15 | 21.55 | 25.16 | 26.96 | 24.38 | 21.22 | 25.02 | 22.98 | 23.81 | 2.63 |

the hemisphere and test lights are randomly sampled from the full hemisphere, and (2) *extrapolation*, where training lights are restricted to one side forming a triangular configuration, and test lights are sampled from the opposite side. In the fixed-elevation-light setting, all training lights are placed at a fixed elevation of $45°$, uniformly distributed along a horizontal ring, while test lights are drawn from elevation bands outside this range (i.e., $\leq 30°$ or $\geq 55°$), simulating fixed-height capture trajectories. We report average results on three synthetic scenes in Table 5. Our method consistently achieves strong performance across various lighting settings. In contrast, both baselines degrade significantly under sparse or biased lighting. In particular, NRHints fails to reconstruct the Ball scene under the Three Lights setting, producing a collapsed geometry, which is similar to the failure case described in the paper's OOD setting. These results demonstrate that MetaGS not only generalizes well to unseen lighting conditions, but also remains robust under extremely limited or clustered illumination. This highlights its practical applicability in real-world relighting scenarios where lighting is sparse, biased, or expensive to capture.

**Environment map generalization.** Our model generalizes effectively to environment map relighting. We approximate global illumination by importance sampling point lights from the environment map based on their pixel intensities. Notably, for our model in particular (unlike the compared methods), the sampled light directions at test time may follow a distribution different from those encountered during training, presenting a significant challenge for OOD generalization. Instead of assuming infinite light source distance, we simulate distant lighting by placing point lights along environment directions at a fixed distance (twice the scene radius). We compare our method with R3G [8] and GaussianShader [9]. Both methods fail to converge when trained directly on out-of-distribution (OOD) data, so we construct training data that covers the full hemispherical illumination space. Even with this setup, GaussianShader still fails to converge, while Relightable 3DGS struggles with the OLAT learning task and exhibits noticeable artifacts. As shown in Figure 6, MetaGS produces diverse and visually plausible relighting results under complex environment maps, demonstrating its robustness and generalization capability.

To further validate MetaGS's capability in handling complex lighting conditions, we conduct experiments with test sets containing 2–3 light sources. We also present the results for free-viewpoint relighting with in-distribution point light positions. Please refer to the Appendix for details.

### 4.4 Meta-Learning with Different Relighting Models

The proposed meta-learning scheme is inherently applicable to other rendering frameworks. To evaluate the generality of the proposed meta-learning scheme, we integrate it into several alternative relighting models across diverse families, including both NeRF-based and 3DGS-based methods,

and spanning from BRDF-based to fully implicit formulations, including NRHints [34], GS$^3$ [1] and RNG [7]. The PSNR results are summarized in Table 6.

While meta-learning consistently enhances OOD generalization across various rendering models, its benefits are most pronounced when paired with explicit geometric modeling and shading methods. Notably, the meta-learning approach yields the largest improvement on our Gaussian-Phong rendering model. This is because the Phong model's simple yet physically grounded design introduces strong inductive biases that enable meta-learning to capture generalized scene structural information even under limited lighting conditions.

## 5 Related Work

Recent differentiable volume rendering techniques, including NeRF-based [17], SDF-based [32, 33], and 3DGS-based [10] methods, have significantly improved the quality and efficiency of novel view synthesis for 3D scenes. NeRF-based methods utilize deep neural networks to model volumetric scene functions, encoding color and density to synthesize high-quality images from sparse viewpoints [17, 15, 37]. However, these approaches typically require substantial training time. 3DGS significantly reduces both training and inference time. It converts point cloud data into a continuous volumetric representation by applying Gaussian kernels to point cloud data, facilitating rendering and further processing. This technique is now widely adopted for efficient 3D and 4D reconstructions across varied data types [28, 9, 16, 26, 31].

The task of 3D relighting involves altering the illumination in a 3D scene while maintaining its geometry. It requires decomposing materials and lighting of a scene from multiple images, which is challenging due to its high-dimensional nature. Recent advances in volume rendering have introduced various solutions for this task, including those based on neural fields [39, 27, 22, 20, 21, 13, 23, 22, 30] and the methods based on Gaussian point clouds [28, 9, 8, 36]. A limitation of these methods is that they require a substantial volume of multi-view images captured under individual lighting conditions, making them impractical in real-world scenarios. Several studies have proposed NeRF-based 3D relighting techniques within the OLAT framework [29, 39, 34], which greatly reduce the demands on training data. Nonetheless, these techniques tend to be computationally intensive because of NeRF's inherent complexity in the volume rendering process. In parallel, concurrent 3DGS-based approaches [6, 1] have also explored the OLAT relighting challenge. Despite their efforts, these approaches still face difficulties when dealing with lighting conditions not encountered during training, similar to the limitations observed in NeRF-based methods.

In contrast, our method addresses this issue by incorporating specific physical priors, represented by the Phong reflection model, into the Gaussian splatting framework. We have also designed a meta-learning approach to enhance the method's generalizability to OOD relighting scenarios.

## 6 Conclusions and Limitations

In this paper, we explored a novel and challenging problem: out-of-distribution (OOD) 3D relighting. To address this, we proposed MetaGS, which builds on Gaussian splatting and presented a novel bilevel optimization-based meta-learning framework that explicitly promotes generalizable Gaussian geometry and appearance learning. This meta-learning formulation allows the model to adapt to varying light sources and viewpoints, even when the training data is biased or sparsely sampled in the lighting space. Furthermore, MetaGS incorporates a differentiable Blinn-Phong reflection model within Gaussian splatting, effectively disentangling lighting effects into ambient, diffuse, and specular shading components, thereby improving the physical realism and reconstruction fidelity under diverse lighting. Extensive experiments on both synthetic and real-world datasets demonstrate that MetaGS significantly outperforms existing OLAT relighting approaches under OOD relighting conditions. It also achieves strong generalization to complex environment maps.

Despite these advances, several limitations remain. An open challenge is ensuring the robustness of our approach under more complex lighting conditions. Moreover, our current framework only accounts for direct illumination; incorporating indirect lighting could further enhance the quality of relighting. Additionally, the current model assumes simple Phong reflection, which may limit fidelity when modeling materials with strong subsurface scattering or anisotropic reflectance.

## Acknowledgements

This work was supported by the National Natural Science Foundation of China (Grant 62250062), the Shanghai Municipal Science and Technology Major Project (Grant 2021SHZDZX0102), and the Fundamental Research Funds for the Central Universities.

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
