# OpenReview forum: "MetaGS: A Meta-Learned Gaussian-Phong Model for Out-of-Distribution 3D Scene Relighting"
_NeurIPS.cc/2025/Conference — NeurIPS 2025 spotlight_

### Official Review · Reviewer_v8Ud · 2025-07-01

**Clarity:** 3
**Significance:** 3
**Originality:** 3
**Rating:** 5
**Confidence:** 4

**Summary:**

The paper presents the MetaGS model, a meta-learning approach for improving out-of-distribution 3D GS scene relighting.
In the task, the 3D object's is first captured One Light At a Time (OLAT), in other words, the light source and camera positions vary at the same time. OLAT is a convenient capture setup, but lighting and geometry is more entangled. Prior work often fails to truly uncover the material / lighting interaction, and as a result leads to poor generalizability under novel lighting condition.

The method tackles this issue using two primary ways:
1. It employs a meta-learning framework using bilevel optimization, which treats each capture sequence as an individual learning task, and use an existing meta learning framework to initialize and update the learnable parameters.
2. It integrates physical priors from the Blinn-Phong model to decouple shading components. This makes the GS rendering model more physically based and more constrained, avoiding overfitting.

**Questions:**

While the meta learning framework is novel and effective, I wish the authors did more experiments on the effect of the Blinn-Phong model. Namely, can you still achieve comparably high OOD test performance if you used another rendering model from prior work? The reason being Blinn-Phong model is still quite restrictive in terms of shadow effect. Neural Relightable Gaussians and several other work learned a MLP decoder instead of physically-based and interpretable material parameters. I'm curious if Blinn-Phong model is a contribution or a limiting factor.

**Ethical Concerns:**

["NO or VERY MINOR ethics concerns only"]

**Final Justification:**

The new experiments in the author's rebuttal excellently answered my concerns. Please add it to the main script.

**Limitations:**

The main limitation seems to be the expressiveness of the Blinn-Phong model.
Currently the authors claimed contribution for the use of this model, however, it's not clear from experiments how and if at all this simplistic modeling actually improves the optimization result.
I'm curious if the meta learning itself is robust enough or it has to be coupled with this Blinn-Phong model.
Some experiments to explore the two contributions independently would clarify my question.

**Quality:**

3

**Strengths And Weaknesses:**

- High impact: the paper tackles a very common challenge in a highly relevant application, relightable GS asset.
- The proposed solution with meta training seems very effective, yielding large PSNR gains in difficult OOD test cases.

---

> ### Author Rebuttal · Authors · 2025-07-30
>
> We thank the reviewer for the comments and note that the main concern lies in the effectiveness of the Blinn-Phong model used in our rendering pipeline. Specifically, whether it remains necessary given the presence of an effective meta-learning framework.
>
> We argue that both the meta-learning framework and the Blinn-Phong shading model play essential and complementary roles in MetaGS, and that their combination is key to achieving strong generalization under OOD lighting. Below, we provide further results and discussion to clarify the respective contributions of the meta-learning component and the Gaussian-Phong renderer.
>
> #### 1. Meta-Learning with Different Relighting Models
>
> > *Can you still achieve comparably high OOD test performance if you used another rendering model from prior work? ...I'm curious if the meta learning itself is robust enough or it has to be coupled with this Blinn-Phong model.*
>
> To investigate the general applicability of our meta-learning method, we incorporate it into several alternative relighting frameworks across different model families—including both NeRF-based and 3DGS-based methods, and ranging from BRDF-based to fully implicit formulations:
> * **Meta-learning + NRHints** (implicit geometry + implicit lighting): This method does not rely on any parametric BRDF model, instead modeling full light transport via an MLP.
> * **Meta-learning + Neural Relightable Gaussians**$^{[1]}$ (explicit geometry + implicit lighting): Each Gaussian is assigned a learned feature vector decoded into color via an MLP (completely free of explicit shading assumptions). *We apply meta-learning in both of its two stages*.
> * **Meta-learning + GS$^3$** (explicit geometry + explicit lighting with MLPs): This method uses a simplified BRDF formulation, along with an MLP-based shadow refinement module, and a residual MLP to compensate for unmodeled lighting effects.
>
> The PSNR results are summarized below (*NaN* indicates model failure during RNG’s second stage):
>
>
> | Method        |        Ball         |       PlasCup       |       RubCup        |         Cat         |      Catsmall       |      CupFabric      |        Fish         |      FurScene       |       Pikachu       |        Pixiu        |        Avg.         | $\Delta$ (Meta$-$Base) |
> |:------------- |:-------------------:|:-------------------:|:-------------------:|:-------------------:|:-------------------:|:-------------------:|:-------------------:|:-------------------:|:-------------------:|:-------------------:|:-------------------:|:----------------------:|
> | **Ours**      |      **26.76**      |      **27.54**      | $\underline{27.95}$ |      **26.45**      |      **26.44**      |      **27.29**      |      **24.68**      |      **24.82**      |      **25.54**      |      **25.65**      |      **26.31**      |           -            |
> | NRHint     |        17.25        |        23.92        |        27.44        |        18.04        |        24.63        |        24.65        |        22.57        |        21.55        |        24.00        |        23.03        |        22.71        |           -            |
> | NRHint + meta |        17.70        | $\underline{24.57}$ |      **28.18**      |        19.75        | $\underline{25.44}$ |        25.83        |        24.04        | $\underline{21.90}$ |        24.51        | $\underline{24.18}$ |        23.61        |          0.90          |
> | RNG       |        20.22        |        22.72        |        24.94        |        *NaN*        |        24.20        |        25.09        |        *NaN*        |        20.81        |        23.55        |        *NaN*        |        23.08        |           -            |
> | RNG + meta    | $\underline{22.65}$ |        23.93        |        25.88        | $\underline{21.94}$ |        25.07        |        26.52        |        23.66        |        21.75        |        24.90        |        22.75        |        23.91        | 1.31 (excluding *NaN*) |
> | GS$^3$     |        18.84        |        20.30        |        24.37        |        17.66        |        23.34        |        25.04        |        21.12        |        17.34        |        24.11        |        19.63        |        21.18        |           -            |
> | GS$^3$ + meta |        21.54        |        23.18        |        26.15        |        21.55        |        25.16        | $\underline{26.96}$ | $\underline{24.38}$ |        21.22        | $\underline{25.02}$ |        22.98        | $\underline{23.81}$ |          2.63          |
>
>
>
> These results demonstrate that while meta-learning consistently improves OOD generalization across different rendering models, its effect is especially pronounced when combined with explicit geometric modeling and shading methods, and is most significant with our Gaussian-Phong formulation.
>
>
>
> [1] Fan, Jiahui, et al. "Rng: Relightable neural gaussians." Proceedings of the Computer Vision and Pattern Recognition Conference. 2025.
>
>
> #### 2. Why the Blinn-Phong Model Works Well with Meta-Learning
> > *The reason being Blinn-Phong model is still quite restrictive in terms of shadow effect... I'm curious if Blinn-Phong model is a contribution or a limiting factor.*
>
> From the above results, we observe that while the meta-learning scheme is compatible with various model designs, its benefits are most pronounced when combined with the physically structured Blinn-Phong model. Below, we outline two key reasons why this combination performs best:
> - **Simplicity \& stable convergence:** Despite its simplicity, the Blinn-Phong model defines a low-dimensional parameter space (ambient, diffuse, specular) for meta-learning optimization. Since meta-learning aims to capture shared structures across tasks (e.g., different lighting conditions in our case), this compact parameterization facilitates smoother and more stable gradient flow during meta-training, resulting in faster convergence and improved generalization of the learned scene geometries.
> - **Straightforward inductive bias of disentanglement:** The physically grounded formulation naturally encourages a separation of global scene representations (e.g., geometry, materials) from lighting-specific illumunation factors. Compared to neural-based alternatives such as **Neural Relightable Gaussians (RNG)**, the Gaussian-Phong model injects an explicit physical prior into the rendering process, constraining illumination components to follow interpretable, physics-based rules. Rather than embedding all lighting effects within a high-dimensional latent space, our method promotes the disentanglement of shared global attributes (e.g., scene geometry and material properties) from lighting-specific factors (e.g., direction and intensity) during meta-training.
>
> It is worth noting that the full potential of the Gaussian-Phong formulation is only unlocked through meta-learning. While its low-dimensional structure provides strong interpretability and regularization, it does come at the cost of expressiveness, as pointed out by the reviewer. When trained in isolation (see our ablation in Table 3, “w/o meta-learning”), the Phong model can indeed lead to suboptimal results. The bilevel meta-learning loop addresses this by validating each inner-loop update on held-out lighting directions, effectively turning the strong physical prior of Phong into a practical advantage in OOD generalization.
>
> #### 3. Summary: A Balanced Trade-Off
> In summary, OOD relighting poses a fundamental trade-off between in-domain expressiveness and OOD generalization. Our experiments demonstrate that combining a lightweight, physically grounded rendering model with meta-learning strikes this balance well. While the proposed Gaussian-Phong model is not the most physically accurate model, it functions as both a reasonably expressive renderer and a compact, differentiable prior that facilitates robust generalization through meta-learning across diverse lighting conditions.

---

> ### Author Response · Authors · 2025-08-04
>
> Dear Reviewer,
>
> Thank you once again for reviewing our paper. We would greatly appreciate it if you could take a moment to review our feedback and newly added experiments, and let us know if any concerns remain.
>
> Best regards,
>
> The Authors

---

> > ### Comment · Reviewer_v8Ud · 2025-08-05
> > **thank you for the response**
> >
> > The new experiments reviewed that the meta-learning works in other rendering models, which was my main concern. I'd like to raise the score based on this new evidence.

---

> > > ### Author Response · Authors · 2025-08-06
> > >
> > > Dear reviewer,
> > >
> > > Thank you for your positive and constructive feedback. Your comments have been instrumental in improving our work, and we greatly appreciate your time and insightful suggestions.
> > >
> > > Best regards,
> > >
> > > The Authors

---

### Official Review · Reviewer_N4oh · 2025-07-02

**Clarity:** 3
**Significance:** 3
**Originality:** 3
**Rating:** 5
**Confidence:** 3

**Summary:**

The paper introduces MetaGS, a novel method for out-of-distribution (OOD) 3D scene relighting, which addresses the challenge of generating realistic novel views under unseen lighting conditions. Existing relighting methods often fail in OOD scenarios due to overfitting to training lighting distributions. MetaGS proposes two key innovations: a meta-learning framework and a differentiable Blinn-Phong model to tackle this issue. Evaluations on synthetic and real-world datasets show MetaGS outperforms baselines in OOD relighting, supports efficient point-light relighting, and generalizes to unseen environment maps.

**Questions:**

The experimental results obtained from 3DGS exhibit unexpectedly and unreasonably poor performance. Notably, given that the output of 3DGS is in the format of OLAT, it is surprising that even a straightforward convolution operation applied to any novel environment map would yield a significantly superior outcome compared to the results presented in the paper.

**Ethical Concerns:**

["NO or VERY MINOR ethics concerns only"]

**Final Justification:**

The authors' response has adequately addressed my concerns about the potential limitations imposed by the employed Blinn-Phong model and the unexpectedly poor performance of GS3.

**Limitations:**

Yes.

**Paper Formatting Concerns:**

No.

**Quality:**

3

**Strengths And Weaknesses:**

**Strengths**
1. OOD Generalization. MetaGS explicitly addresses the OOD relighting challenge, where test lighting conditions differ from training. Its meta-learning scheme simulates unseen lighting during training, reducing overfitting and improving generalization to novel illuminations.
2. Physical Priors Integration. By incorporating the Blinn-Phong model, MetaGS decouples shading components (ambient, diffuse, specular), enabling more accurate and physically grounded relighting. This structural decomposition enhances the model’s ability to adapt to new lighting scenarios.

**Weaknesses**
The incorporation of parametric material modeling, like the Blinn-Phong model used here, is not a judicious approach in this context. In contrast, 3DGS is a potent and universally applicable 3D representation capable of accommodating any material theoretically. Notably, the adoption of parametric models will inevitably impose limitations on the representational capabilities of 3DGS.

---

> ### Author Rebuttal · Authors · 2025-07-31
>
> We appreciate the reviewer’s time and effort in reviewing our manuscript and have responded to the points raised as follows.
>
> > *Q1. The adoption of parametric models, like Blinn-Phong model, will inevitably impose limitations on the representational capabilities of 3DGS.*
>
> We thank the reviewer for raising this question. This touches on a fundamental trade-off between expressiveness and generalization ability. While we agree with the reviewer that introducing parametric reflectance models like Blinn-Phong may limit the representational capabilities of 3DGS, our design choice is motivated by the need to improve generalization, particularly in OOD scenarios.
>
> We fully acknowledge that 3DGS is, in theory, capable of modeling arbitrary appearance and reflectance. However, in practice, especially under limited and biased OLAT data, standard 3DGS comes with significant risk of **overfitting to training light distributions** and **entangled specular/diffuse representations**, as we observe in our baseline comparisons (e.g., GS$^3$, WildLight).
>
> The goal of incorporating a simple parametric model like Blinn-Phong is not to improve the rendering quality of 3DGS under rich and fixed lighting conditions, but to introduce a **lightweight, physically grounded inductive bias** that encourages the disentanglement of ambient, diffuse, and specular reflections, thereby enabling more robust and interpretable modeling of light transport.
>
> In OOD relighting tasks, generalization is often more crucial than raw representational power. Despite its simplicity, the Phong model offers physically meaningful priors that help constrain the learning space under sparse and varied lighting conditions. Importantly, this inductive bias synergizes with our meta-learning algorithm, promoting the separation of global scene factors (e.g., geometry and material properties) from lighting-specific variables (e.g., direction and intensity). This significantly enhances generalization performance, as demonstrated in our experiments.
>
> In summary, while we agree that parametric models like Blinn-Phong impose certain representational constraints, we believe that in the context of OOD relighting, these constraints serve as useful inductive priors. Our work takes a first step toward integrating physically inspired models into 3DGS to improve generalization, and we see great potential in future work exploring more powerful models that retain these desirable physical properties.
>
>
>
> > *Q2. The experimental results obtained from 3DGS exhibit unexpectedly and unreasonably poor performance. Notably, given that the output of 3DGS is in the format of OLAT, it is surprising that even a straightforward convolution operation applied to any novel environment map would yield a significantly superior outcome compared to the results presented in the paper.*
>
> We appreciate the reviewer's comments and are committed to addressing all concerns thoroughly. However, we would like to kindly ask for some clarification to ensure we fully understand the points raised.
>
> Could you please specify which particular experiment or result you refer to when mentioning the "*unexpectedly poor performance*" of 3DGS? Additionally, we are unclear about the phrase "*a straightforward convolution operation applied to any novel environment map*". Could you please elaborate on what this refers to? For example, is this referencing a known baseline or a particular technique?
>
> Your further explanation will greatly assist us in providing a precise and constructive response. We thank you in advance for your guidance. We are also happy to provide a more detailed explanation during the subsequent author-reviewer discussion phase.

---

> > ### Comment · Reviewer_N4oh · 2025-08-05
> >
> > Thanks to the authors for their detailed response. I would like to correct a typo in my initial comments: I intended to question the poor performance of GS3, not 3DGS.

---

> > > ### Author Response · Authors · 2025-08-05
> > >
> > > We thank the reviewer for the helpful clarifications and the opportunity to elaborate on this point.
> > >
> > > We agree with the reviewer’s observation that GS$^3$--despite producing OLAT-formatted outputs--fails significantly under our OOD relighting setup. This failure can be primarily attributed to GS$^3$'s high reliance on flexible yet weakly-constrained components, such as the MLP-based shadow refinement module and the residual MLP designed to approximate unmodeled lighting effects. While such modules provide strong representational capacity and fitting power, they are prone to memorizing the biased illumination patterns present in limited OLAT data, rather than capturing the underlying principles of light transport. As a result, GS$^3$ often learns *spurious correlations* between lighting and appearance, which do not generalize to unseen lighting conditions. In our visualizations, we observe visually implausible behaviors such as erroneous specular highlights and incorrect shadow placements. For instance, in the Plastic Cup scene (Fig. 3), the model casts unexpected dark regions in areas that are directly illuminated, or introduces isolated specular glints where no such effects should occur. These artifacts suggest that the learned representation may entangle shading with visibility or unhandled effects in an inconsistent manner.
> > >
> > > In contrast, our method leverages a meta-learning framework to encourages the learning of light-independent geometry and appearance attributes, resulting in more interpretable and physically consistent rendering under out-of-distribution illuminations. The synergy between physical priors and generalization-driven optimization enables our approach to produce robust and physically coherent relighting results, even under severe distribution shifts where GS$^3$ fails to generalize.

---

> > > > ### Comment · Reviewer_N4oh · 2025-08-06
> > > >
> > > > I appreciate the authors' comprehensive response. My concerns regarding the potential limitations imposed by the employed Blinn-Phong model and the unexpectedly poor performance of GS3 have been addressed. Hence, I decide to raise the score accordingly. I recommend that the authors incorporate these additional analyses into the revision.

---

> > > > > ### Author Response · Authors · 2025-08-07
> > > > >
> > > > > Dear reviewer,
> > > > >
> > > > > Thank you for your positive and constructive feedback. We will carefully incorporate the additional analyses into our revision. Your comments have been instrumental in improving our work, and we greatly appreciate your time and insightful suggestions.
> > > > >
> > > > > Best regards,
> > > > >
> > > > > The Authors

---

> ### Author Response · Authors · 2025-08-04
>
> Dear Reviewer,
>
> Thank you once again for reviewing our paper. We would greatly appreciate it if you could take a moment to review our feedback, and let us know if any concerns remain. We would be happy to answer them before the end of the discussion period.
>
>
> Best regards,
>
> The Authors

---

### Official Review · Reviewer_m2RQ · 2025-07-03

**Clarity:** 3
**Significance:** 4
**Originality:** 3
**Rating:** 5
**Confidence:** 4

**Summary:**

This paper tackles the challenging problem of out-of-distribution (OOD) 3D relighting, where the goal is to synthesize novel views under lighting conditions not seen during training. The authors propose MetaGS, a method that combines meta-learning with 3D Gaussian Splatting to improve generalization across diverse lighting environments. The approach encourages the learning of geometry and appearance features that transfer well to unseen lighting, even when the training data is biased. In addition, MetaGS incorporates a differentiable Blinn-Phong reflection model into the Gaussian framework to better disentangle shading components and improve physical realism. Experiments on both synthetic and real-world datasets show that MetaGS consistently outperforms existing methods in OOD relighting scenarios, demonstrating strong generalization to new light sources, including complex environment maps.

**Questions:**

- As mentioned in the weaknesses, it would be helpful to know whether the proposed training framework can be applied to other existing relighting models. If so, showing the results of such experiments could further support the claim that the meta-learning approach is model-agnostic and broadly applicable.

- According to line178, the implementation of MetaGS is done in PyTorch. Is there a specific reason why CUDA was not used for performance optimization? A brief explanation would clarify whether there are any trade-offs or constraints involved in the current implementation.

**Ethical Concerns:**

["NO or VERY MINOR ethics concerns only"]

**Final Justification:**

The authors have effectively addressed my concern regarding the generalizability of the proposed meta-learning-based method to other models, providing appropriate experimental results. Therefore, I will maintain my initial rating of accept.

**Limitations:**

Yes.

**Paper Formatting Concerns:**

I found no major formatting issues.

**Quality:**

4

**Strengths And Weaknesses:**

> Strengths
- The paper is well-organized with a clear and logical flow, making it easy to follow.
- Tackling a more challenging relighting scenario through a multi-task learning approach seems effective.
- While the main focus is on the OLAT relighting problem, the method also shows strong performance under environment map-based relighting, highlighting its generalization capabilities.
- The proposed method consistently outperforms existing baselines, both quantitatively and qualitatively.

> Weaknesses
- One of the core contributions, the meta-learning scheme, feels relatively general and could potentially be applied to other relighting models as well. While the implementation details may differ, showing that this training paradigm can benefit other architectures would have made the case for MetaGS being model-agnostic even stronger.

---

> ### Author Rebuttal · Authors · 2025-07-30
>
> We appreciate the reviewer’s insightful questions and provide our responses below.
>
>
> > *Q1. Apply meta-learning to other relighting models.*
>
> Yes, the proposed meta-learning scheme is inherently applicable to other rendering frameworks. To evaluate its generality, we integrate it into several alternative relighting models across diverse families, including both NeRF-based and 3DGS-based methods, and spanning from BRDF-based to fully implicit formulations:
> * **Meta-learning + NRHints** (implicit geometry + implicit lighting): This method does not rely on any parametric BRDF model, instead modeling full light transport via an MLP.
> * **Meta-learning + Neural Relightable Gaussians**$^{[1]}$  (explicit geometry + implicit lighting): Each Gaussian is assigned a learned feature vector decoded into color via an MLP (completely free of explicit shading assumptions). *We apply meta-learning in both of its two stages*.
> * **Meta-learning + GS$^3$** (explicit geometry + explicit lighting with MLPs): This method uses a simplified BRDF formulation, along with an MLP-based shadow refinement module, and a residual MLP to compensate for unmodeled lighting effects.
>
>
>
> The PSNR results are summarized below (*NaN* indicates model failure during RNG’s second stage):
>
> | Method        |        Ball         |       PlasCup       |       RubCup        |         Cat         |      Catsmall       |      CupFabric      |        Fish         |      FurScene       |       Pikachu       |        Pixiu        |   Avg.    |
> |:------------- |:-------------------:|:-------------------:|:-------------------:|:-------------------:|:-------------------:|:-------------------:|:-------------------:|:-------------------:|:-------------------:|:-------------------:|:---------:|
> | **Ours**      |      **26.76**      |      **27.54**      | $\underline{27.95}$ |      **26.45**      |      **26.44**      |      **27.29**      |      **24.68**      |      **24.82**      |      **25.54**      |      **25.65**      | **26.31** |
> | NRHint    |        17.25        |        23.92        |        27.44        |        18.04        |        24.63        |        24.65        |        22.57        |        21.55        |        24.00        |        23.03        |   22.71   |
> | NRHint + meta |        17.70        | $\underline{24.57}$ |      **28.18**      |        19.75        | $\underline{25.44}$ |        25.83        |        24.04        | $\underline{21.90}$ |        24.51        | $\underline{24.18}$ |   23.61   |
> | RNG       |        20.22        |        22.72        |        24.94        |        *NaN*        |        24.20        |        25.09        |        *NaN*        |        20.81        |        23.55        |        *NaN*        |   23.08   |
> | RNG + meta    | $\underline{22.65}$ |        23.93        |        25.88        | $\underline{21.94}$ |        25.07        |        26.52        |        23.66        |        21.75        |        24.90        |        22.75        |   23.91   |
> | GS$^3$    |        18.84        |        20.30        |        24.37        |        17.66        |        23.34        |        25.04        |        21.12        |        17.34        |        24.11        |        19.63        |   21.18   |
> | GS$^3$ + meta |        21.54        |        23.18        |        26.15        |        21.55        |        25.16        | $\underline{26.96}$ | $\underline{24.38}$ |        21.22        | $\underline{25.02}$ |        22.98        |   23.81   |
>
>
> These results show that while meta-learning consistently enhances OOD generalization across various rendering models, its benefits are most pronounced when paired with explicit geometric modeling and shading methods. Notably, the meta-learning approach yields the largest improvement on our Gaussian-Phong rendering model. This is because the Phong model’s simple yet physically grounded design introduces strong inductive biases that enable meta-learning to capture generalized scene structural information even under limited lighting conditions.
>
> [1] Fan, Jiahui, et al. "Rng: Relightable neural gaussians." Proceedings of the Computer Vision and Pattern Recognition Conference. 2025.
>
> > *Q2. Is there a specific reason why CUDA was not used for performance optimization?*
>
> Thank you for your question. Our implementation is fully GPU-accelerated and leverages PyTorch's built-in CUDA operations throughout. We did consider writing custom CUDA kernels, but found that the built-in operations were already highly optimized and sufficient to meet our performance needs. Moreover, our rasterization pipeline builds directly upon the same CUDA-accelerated primitives used in 3DGS, which are sufficient to achieve efficient training. As such, introducing additional low-level kernels would have increased implementation complexity without yielding significant performance gains in our setting.

---

> > ### Comment · Reviewer_m2RQ · 2025-08-06
> >
> > I would like to thank the authors for their thoughtful and well-prepared rebuttal. The additional experiments demonstrate the applicability of the proposed method to other models and show promising results. The authors have adequately addressed the concerns I raised in my initial review, and I will maintain my initial rating.

---

> > > ### Author Response · Authors · 2025-08-06
> > >
> > > Dear reviewer,
> > >
> > > Thank you for your positive and constructive feedback. Your comments have been instrumental in improving our work, and we greatly appreciate your time and insightful suggestions.
> > >
> > > Best regards,
> > >
> > > The Authors

---

> ### Author Response · Authors · 2025-08-04
>
> Dear Reviewer,
>
> Thank you once again for reviewing our paper. We would greatly appreciate it if you could take a moment to review our feedback and newly added experiments, and let us know if any concerns remain.
>
> Best regards,
>
> The Authors

---

### Official Review · Reviewer_Wg3c · 2025-07-05

**Clarity:** 3
**Significance:** 2
**Originality:** 2
**Rating:** 4
**Confidence:** 4

**Summary:**

This paper introduces MetaGS, a novel method for 3D scene relighting under out-of-distribution (OOD) lighting conditions using 3D Gaussian Splatting. Unlike previous OLAT (One-Light-At-a-Time) methods that struggle with generalization to unseen lighting, MetaGS addresses this by combining a differentiable Blinn-Phong reflection model to disentangle shading components and a bilevel meta-learning framework to learn lighting-independent scene representations. The approach enhances the model's ability to generalize across varying lighting conditions, even with spatially biased training data, and shows strong performance on both synthetic and real-world datasets without requiring retraining for new lighting distributions.

**Questions:**

1. The paper doesn’t clarify how many distinct light positions are used during training. To better assess OOD robustness, could the authors evaluate performance when training with very few or clustered lights? This would test the method under more realistic and challenging conditions. Strong results in such cases would significantly strengthen the paper’s impact.
2. Since OLAT images are captured under individual point light sources, it is feasible to synthesize ground truth renderings for environment map relighting by linearly combining OLAT images according to the lighting distribution in the environment map. However, the paper provides only qualitative results for this experiment, without reporting any quantitative metrics or comparisons with ground truth images. Including such ground truth-based evaluations (e.g., PSNR, SSIM, LPIPS) would significantly strengthen the claim of generalization to complex lighting. Could the authors provide comparisons with ground truth images to further validate the results?

**Ethical Concerns:**

["NO or VERY MINOR ethics concerns only"]

**Final Justification:**

The rebuttal, along with the additional experiments, has alleviated many of my concerns to a certain extent.

**Limitations:**

yes

**Quality:**

3

**Strengths And Weaknesses:**

Strengths:
1. MetaGS effectively tackles the challenge of out-of-distribution (OOD) lighting in 3D relighting. By leveraging a bilevel meta-learning strategy, the model simulates unseen lighting conditions during training, enabling it to learn lighting-invariant geometry and reflectance attributes. This allows robust relighting under novel lighting setups without retraining.
2. The method incorporates a differentiable Blinn-Phong reflection model into Gaussian splatting, introducing physically grounded priors. This enables the model to explicitly disentangle ambient, diffuse, and specular components, enhancing the realism and interpretability of relighting results.

Weaknesses:
1. This work focuses on out-of-distribution (OOD) lighting scenarios from biased OLAT data. However, real OLAT captures use a single camera with one moving point light, limiting lighting diversity and making extreme OOD cases rare in practice. Thus, the OOD problem here is largely artificial and not representative of typical real-world conditions.

---

> ### Author Rebuttal · Authors · 2025-07-31
>
> We thank the reviewer for the valuable comments and address them below.
>
> >  *Q1. The OOD problem here is largely artificial and not representative of typical real-world conditions.*
>
> We appreciate the reviewer's comment regarding the nature of the OOD lighting setup in our work. While we agree that some OLAT setups (e.g., using a handheld flashlight) may not always exhibit extreme OOD lighting conditions, we respectfully clarify that our OOD lighting definition is not artificial, but rather a practical abstraction of real-world limitations. We address this concern from three perspectives:
>
> #### 1. Real-world OLAT captures are often spatially constrained
> Even with a mobile light source, physical constraints (e.g., tripod positioning, occlusions, user-defined capture paths) typically result in non-uniform and biased light distributions. For example, OLAT captures using smartphone flashlights (like in [5, 34]) or mounted light rigs are frequently constrained to a subset of hemisphere directions. Unlike controlled lab environments, real-world conditions often involve relighting from previously unobserved directions (e.g., when user illumination changes or devices rotate). In such cases, generalization to unseen or poorly sampled lighting directions becomes necessary, which is exactly the scenario our OOD setup simulates.
>
> Therefore, our OOD lighting split is designed to test the generalization capability of relighting models, which is a crucial requirement for practical deployment. As shown in Fig. 1, 3, and 4, existing methods suffer noticeable degradation even under moderate lighting distribution shifts, while MetaGS generalizes well.
>
> #### 2. OOD OLAT is only a training/test protocol; Generalization is the goal
> While we train on OLAT, our method evaluates under broader settings, including environment map relighting and camera-light-colocated setups. These results (Tab. 4, Fig. 6) demonstrate that our method's benefits are not limited to synthetic OOD cases but also transfer to practical test-time distributions beyond the training configuration.
>
> #### 3. Broader applications
> To better simulate realistic lighting biases, we conduct additional experiments restricting training lights to a fixed elevation, mimicking a planar circular light path, while the testing lights are placed at different heights. We also evaluate it under OOD clustered-light and few-light training setups, as detailed in our **Responses to Q2**. These experiments further support the method's applicability beyond synthetic extremes.
>
> Finally, we would like to discuss another practical scenario--**OLAT relighting for dynamic scenes**. In such settings, where a scene is captured by a single camera and illuminated by a single light source (common in casual or consumer-grade setups), the scene’s motion is typically non-reproducible, preventing the dense sampling of illumination conditions. As a result, illumination coverage is inherently sparse over time, introducing a severe OOD relighting problem. While our current work does not explicitly target dynamic scenes, this scenario highlights the real-world significance of our OOD formulation. We see this as a promising direction for future work.
>
> We thank the reviewer again for raising this important concern and hope this clarifies our motivation and rationale behind the OOD setup.
>
>
> > *Q2. (1) The paper doesn’t clarify how many distinct light positions are used during training. (2) To better assess OOD robustness, could the authors evaluate performance when training with very few or clustered lights?*
>
> #### 1. Number of light positions
> In our OLAT setup, each image features a distinct light position and a distinct camera viewpoint. Specifically, we use 500 images (i.e., 500 unique point-light positions) for each synthetic scene and 600 images for each real-world scene. These capture details are provided in Appendix B (Lines 404–406).
>
>
> #### 2. Training with few or clustered lights
> To evaluate MetaGS under constrained illumination conditions, we additionally design three settings that vary in light distribution, coverage, and sparsity, simulating more realistic capture scenarios:
> - **Clustered Lights:** We construct lighting configurations with 1, 2, or 3 clusters on the illumination hemisphere. Each configuration is evaluated under two distinct settings:
>     * *Interpolation lighting:* The cluster centers of the training lights are uniformly distributed across the hemisphere; test lights are placed between clusters.
>     * *Extrapolation lighting:* Training lights are sampled from one side of the hemisphere, with cluster centers uniformly distributed (e.g., for two clusters, their centers share the same elevation of $45^\circ$; for three clusters, they form a triangular configuration). Test lights are randomly sampled from the opposite side.
> - **Few Lights:** We sample 3 training lights in this setting, each paired with 200 camera views (i.e., one-light–multi-camera). Training light positions match the cluster centers used in the *Clustered Lights* setting.
>     * *Interpolation:* Training lights are uniformly distributed; test lights sampled from the full hemisphere.
>     * *Extrapolation:* Training lights are confined to one side; test lights from the opposite side.
> - **Fixed-elevation Lights:** All training lights are placed at a fixed elevation of $45^\circ$, uniformly sampled along a horizontal ring. Test lights are drawn elevation bands outside the training range (i.e., $\leq 35^\circ$ or $\geq 55^\circ$). This simulates fixed-height capture trajectories.
>
> All settings (except Few Lights) use 500 training images and 100 test images rendered under OLAT conditions, with unique light-camera pairs. We report average results on three synthetic scenes (Blender-rendered) in terms of PSNR / SSIM / LPIPS.
>
> Clustered Lights:
>
> |  | Single cluster (Interp) | Single cluster (Extrap) | Two clusters (Interp) | Two clusters (Extrap) | Three clusters (Interp) | Three clusters (Extrap) |
> |---------|:--------------:|:--------------------:|:------------:|:------------------:|:--------------:|:---------------------:|
> | NRHints | 21.14/0.9256/0.0580 | 18.81/0.9049/0.0705 | 22.55/0.9391/0.0512 | 20.45/0.9232/0.0606 | 26.45/0.9773/0.0325 | 22.17/0.9370/0.0540 |
> | GS$^3$  | 19.95/0.9184/0.0663 | 18.32/0.8895/0.0647 | 20.51/0.9286/0.0627 | 19.19/0.9117/0.0680 | 25.17/0.9640/0.0494 | 20.20/0.9223/0.0630 |
> | Ours    | **24.09/0.9350/0.0493** | **23.29/0.9282/0.0539** | **26.13/0.9492/0.0387** | **25.18/0.9417/0.0489** | **28.13/0.9678/0.0316** | **26.85/0.9550/0.0427** |
>
> Few Lights \& Fixed-elevation Lights:
>
> |         | Fixed-elevation | Few lights (Interp) | Few lights (Extrap) |
> |---------|:---------------:|:----------:|:----------------:|
> | NRHints | 25.76/0.9633/0.0402 | 18.05/0.8823/0.0911 | 18.62/0.8960/0.0681 |
> | GS$^3$  | 22.93/0.9393/0.0580 | 19.92/0.9165/0.0693 | 18.06/0.8781/0.0920 |
> | Ours    | **27.48/0.9590/0.0371** | **24.90/0.9476/0.0462** | **23.94/0.9369/0.0533** |
>
> **Observations and insights:** As shown in the above results, our method consistently achieves strong performance across various lighting settings. In contrast, both baselines degrade significantly under sparse or biased lighting (e.g., Single Cluster, Two Clusters, and Few Lights). In particular, NRHints fails to reconstruct the Ball scene under the Three Lights setting, producing a collapsed geometry, which is similar to the failure case described in the paper's OOD setting. These results demonstrate that MetaGS not only generalizes well to unseen lighting conditions, but also remains robust under extremely limited or clustered illumination. This highlights its practical applicability in real-world relighting scenarios where lighting is sparse, biased, or expensive to capture.
>
> > *Q3. Quantitative result of envmap relighting.*
>
> We note that this setting poses a particularly challenging generalization task, as our model is trained exclusively on OLAT data without any access to global illumination or multi-light supervision. While environment map relighting is not the primary target of our design, the results still reflect the model’s capacity to generalize beyond the OLAT setting. To this end, we supplement the following experiments:
>
> **Data construction:** Constructing ground truth via weighted combinations of real OLAT images often leads to errors, due to limited coverage of light directions and colors from importance-sampled environment maps. To ensure reliable evaluation, we generate envmap-illuminated scenes synthetically using Blender, and conduct this comparison only on synthetic scenes.
>
> **Baselines for comparison:** Most OLAT relighting methods (e.g., GS$^3$, NRHints, WildLight) do not support environment relighting. Methods that do support it (e.g., BiGS, PRTGaussian) typically require additional supervision such as all-light-on captures or multi-camera, multi-light setups, which fall outside our assumed constraints. We therefore use Relightable 3DGS as the primary baseline, since it supports environment map rendering under comparable supervision.
>
> |                  | Ball | PlasCup | RubCup |
> |------------------|:----:|:-------:|:------:|
> | Relightable 3DGS | 20.88/0.8432/0.1488 | 22.07/0.8644/0.1294 | 22.15/0.8390/0.1366 |
> | Ours             | **22.70/0.9018/0.0975** | **23.10/0.9335/0.0651** | **23.55/0.9306/0.0736** |
>
> As shown, our method significantly outperforms the baseline in SSIM and LPIPS, with smaller gains in PSNR. We attribute this to imperfect brightness alignment, since environment maps are not seen during training and their HDR intensity values are manually calibrated at test time, potentially introducing global color shifts. Nonetheless, the strong perceptual scores and structural consistency highlight the robustness and generalization capability of our method to novel lighting conditions.

---

> ### Author Response · Authors · 2025-08-04
>
> Dear Reviewer,
>
> Thank you once again for reviewing our paper. We would greatly appreciate it if you could take a moment to review our feedback and newly added experiments, and let us know if any concerns remain.
>
> Best regards,
>
> The Authors

---

> > ### Comment · Reviewer_Wg3c · 2025-08-06
> >
> > I would like to thank the authors for their thoughtful and well-prepared rebuttal.
> > Thanks to their detailed response, I have decided to raise my score.

---

> > > ### Author Response · Authors · 2025-08-06
> > >
> > > Dear reviewer,
> > >
> > > Thank you for your positive and constructive feedback. Your comments have been instrumental in improving our work, and we greatly appreciate your time and insightful suggestions.
> > >
> > > Best regards,
> > >
> > > The Authors

---

### Note · Authors · 2025-08-12

Dear Reviewers, Area Chairs, Senior Area Chairs and Program Chairs,

We sincerely thank you for your time and effort in reviewing our work. During the rebuttal and discussion phases, we carefully addressed all questions and concerns raised by the reviewers, supplementing our responses with additional experiments. Specifically, we report two new sets of experiments:

1. Broader OOD Lighting Settings -- We evaluated our method under more diverse out-of-distribution scenarios (e.g., clustered lights, few lights). The results demonstrate that our approach maintains strong performance in constrained lighting distributions, whereas baseline methods degrade sharply under sparse or biased lighting.


2. Meta-Learning across Relighting Models -- We integrated our meta-learning scheme into various relighting frameworks to assess its generalization benefits. The results show that meta-learning consistently improves OOD generalization across different rendering models, with the largest gains observed for our Gaussian-Phong renderer. This further highlights that the combination of the meta-learning framework and the Blinn-Phong shading model is key to achieving strong OOD lighting generalization.

We greatly appreciate the reviewers’ thorough evaluation and insightful comments. The additional experiments and analyses will be carefully incorporated into the revised version of our paper.

Thank you again for your careful consideration and constructive feedback throughout the review process.

Best regards,

The Authors

---

### Decision · Program_Chairs · 2025-09-17

**Decision:**

Accept (spotlight)

**Comment:**

This submission receives 1 borderline accept and 3 accepts. As recognized by the reviewers during the rebuttal discussion period, this submission has several contributions:
1. tackle the challenging task, out-of-distribution relighting of 3d scenes.
2. propose a novel meta-learning paradigm that is agnostic to the specific relighting model, leading to general improvements.
3. the proposed method consistently outperforms existing baselines both quantitatively and qualitatively, yielding large PSNR gains in difficult OOD test cases.

Although the method is only evaluated at asset level rather than scene level, current evaluation is sufficient and comprehensive, and contribution is significant for an important and challenging task in relevant community. Therefore, the recommendation is acceptance with spotlight.